# Influence of Polyisobutylene Kerosene Additive on Combustion Efficiency in a Liquid Propellant Rocket Engine

Igor Borovik [1,†,*] , Evgeniy Strokach [1,†], Alexander Kozlov [1,†], Valeriy Gaponov [2,†], Vladimir Chvanov [2,†], Petr Levochkin [2,†] and Youngbin Yoon [3,†]

1   Institute No. 2 "Aviation rocket engines and power plants", Moscow Aviation Institute, Volokolamskoe shosse 4, Moscow 125993, Russia; evgenij.strokatsch@mai.ru (E.S.); kozlov202@mai.ru (A.K.)
2   NPO Energomash, Burdenko st. h. 1, Khimki 141400, Russia; gaponov.vd@yandex.ru (V.G.); chvanov_vk@npoem.ru (V.C.); levochkin_ps@npoem.ru (P.L.)
3   Department of Aerospace Engineering, Seoul National University, 1 Gwanak-ro, Gwanak-gu, Seoul 08826, Korea; ybyoon@snu.ac.kr
*   Correspondence: borovik.igor@mai.ru; Tel.: +7-499-158-47-81
†   These authors contributed equally to this work.

**Abstract:** The combustion of kerosene with the polymer additive polyisobutylene (PIB) was experimentally investigated. The aim of the study was to measure the effect of PIB kerosene on the efficiency of combustion chamber cooling and the combustion efficiency of the liquid propellant for a rocket engine operating on kerosene and gaseous oxygen (GOX). The study was conducted on an experimental rocket engine using kerosene wall film cooling in the combustion chamber. Fire tests showed that the addition of polyisobutylene to kerosene had no significant effect on the combustion efficiency. However, analysis of the wall temperature measurement results showed that the use of PIB kerosene is more effective for film cooling than pure kerosene, which can increase the efficiency of combustion chamber cooling and subsequently increase its reliability and reusability. Thus, the findings of this study are expected to be of use in further investigations of wall film cooling efficiency.

**Keywords:** polyisobutylene; drag reduction; kerosene; rocket engine; film cooling; fire test; non-Newtonian liquid

## 1. Introduction

Inclusion of polymers into hydrocarbon fuels is a well-known procedure for hydrodynamical drag reduction. The decrease in hydrodynamic drag is caused by the Toms effect. In 1948, B. Toms [1] recognized the reduction in wall shear stress caused by the addition of small amount of polymer to a turbulent flowing fluid. This effect is widely used in pipeline transportation of oil and shipbuilding to reduce friction. The application of kerosene polymer additives for liquid propellant rocket engine (LPRE) efficiency enhancement has been studied recently. The use of additives may decrease pressure losses in the combustion chamber cooling channels by 20%–24%, enhance fuel pump efficiency by 13%–17%, and decrease turbine power by 7%–9% [2]. Polymer additives are most effective for engines with gas-generator cycles because the polymer additives in kerosene reduce the fuel massflow driving the turbine and do not participate in thrust creation.

It is commonly known that polymer additives influence the process of liquid atomization into droplets, increase the droplet mean diameter (DMD), and alter the droplet diameter distribution [3,4]. This feature of polymer additives is widely used for reduction of fuel mist ignition during fuel tank

destruction in catastrophic events. Therefore, it is obvious to assume that the addition of polymers in kerosene would decrease the combustion performance due to increase in the droplet mean diameter.

However, the change in droplet mean diameter is not the main factor influencing combustion efficiency. In some cases, even a large increase in the DMD does not reduce combustion performance because the droplet diameter distribution type is also a highly important criterion. This is due to the fact that completely different shapes of the droplet diameter distributions can correspond to the same DMD. Also, two injectors spraying a liquid with the same average diameter can provide different combustion efficiency due to the fact that their design will give different droplet size distributions. This is because a different ratio of large and small droplets injected will give different combustion efficiency [5,6].

Estimation of the influence of polyisobutylene (PIB) additives on the combustion process during fire testing is difficult as the polymer additives are often destroyed in the cooling jacket channels and turbopump impeller. As a result, kerosene fuel seeps into the injector containing the destroyed polymer additive molecules. Previously conducted tests show that the use of PIB kerosene additives in modern LPREs (RD-107, RD-58M, RD-170) does not influence the operating process in the combustion chamber [7]. However, if the polymer molecules seep into the injector in the half-destroyed or undestroyed conditions, a different behavior may be seen, as other physical effects can be revealed, such as the drag reduction (main Toms effect), which can affect the flow parameters. This situation is possible during the use of polymer additives containing very large molecules, which in the pump or cooling channels may separate into smaller but still large fragments comparable to the size of the PIB molecules [8,9]. In this case, the polymer additive may significantly affect the combustion performance and eliminate the positive effects of drag reduction in the fuel feeding system.

The combustion process in the combustion chamber of an LPRE is a complicated physical phenomenon, which consists of a number of interdependent processes, namely, fuel flowing out of the injector, fuel jet atomization into droplets and ligaments, fuel heating and evaporation, and ignition and combustion in the gaseous phase.

Previously, several studies were conducted to describe the influence of PIB additive on the combustion process of kerosene in gaseous oxygen [7,10]. These studies, however, did not consider the issue of PIB influence on combustion efficiency in the presence of liquid (namely kerosene) film cooling.

This study aims to continue the previous research and is dedicated to investigation of the influence of PIB additives on the combustion process in the combustion chamber with kerosene film cooling, i.e., reductive conditions of the combustion gas, which is more typical of modern LPREs.

## 2. Objective of Study

The objective of the present study concerns the effects of two types of kerosene: with PIB additives and without additives (pure kerosene). Table 1 describes the basic physical and chemical characteristics of kerosene in comparison with samples containing PIB additive in varying concentrations [11]. The data show that all physical properties of the kerosene samples are almost equal.

**Table 1.** Physical properties of kerosene with various polyisobutylene (PIB) concentrations.

| Physical Properties | Sample Analysis Results | | | |
|---|---|---|---|---|
| | 0.00% by Mass | 0.01% by Mass | 0.05% by Mass | 0.1% by Mass |
| Density at 20 °C, kg/m$^3$ | 835.6 | 835.6 | 835.6 | 835.7 |
| Kinematic viscosity at 23 °C, mm$^2$/s | 2.35 | 2.54 | 3.1 | 3.9 |
| Melting temperature, °C | −60 | −65.0 | −65.1 | −64.9 |
| Lower heat of combustion, kJ/kg | 43,135 | 43,145 | 43,111 | 43,125 |
| Saturated vapor pressure at 37.8 °C, kPa | 5.5 | 5.7 | 6.5 | 6.9 |
| Surface tension at 25 °C, mN/m | 27.7 | 27.4 | 27.4 | 27.4 |
| Lower heat capacity at 20 °C, kJ/(kg·K) | 1.898 | 1.898 | 1.898 | 1.898 |
| Thermal conductivity at 20 °C, W/(m·K) | 0.122 | 0.112 | 0.112 | 0.112 |
| Evaporation heat, kJ/kg | 12.4 | 12.4 | 12.4 | 12.4 |

## 3. Methods

In order to estimate the influence of PIB additive in kerosene on the combustion performance, a special experimental LPRE using gaseous oxygen and kerosene was used in the study (Figure 1). The main feature of this design was a separate inlet for liquid component for the wall film cooling. In this manner, the design allows for regulating the wall film cooling massflow without affecting the combustion chamber core flow.

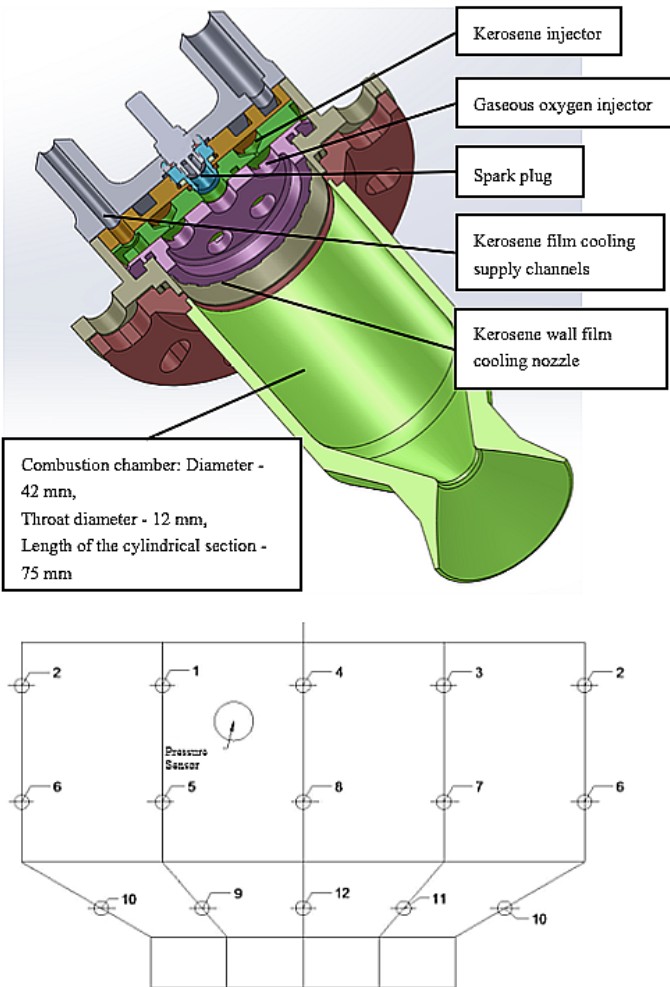

**Figure 1.** Experimental liquid propellant rocket engine (LPRE) schematic and thermocouple (1–12) layout scheme on combustion chamber outer wall.

All fire experiments lasted 2 s. In each experiment, the gaseous oxygen valve was the first to open, after 50 ms the spark plug was switched on, after 50 ms the valve of kerosene injectors opened, after that in 200 ms the spark plug was switched off and in 100 ms the film cooling valve was opened. After that, in 1.7 s, the valve of the kerosene injectors was closed, then after 30 ms the film cooling valve was closed, and the valve of gaseous oxygen was the last to close. Experiments with pure kerosene and kerosene with PIB were conducted in the same sequence. All modes of the experiments are presented in Table 2. Such test conditions were used to reduce the temperature of the hot gas near the wall and to prevent overheating of the combustion chamber wall. Due to the fact that the liquid film under the combustion chamber conditions is atomized by the hot gas flow, the massflow of kerosene for the wall film cooling was chosen so that it was guaranteed to prevent this.

**Table 2.** Experimental results.

| Massflow GOX, in Main Injectors, g/s | Massflow Kerosene in Main Injectors, g/s | Massflow Kerosene in Film Cooling Injectors, g/s | Total Massflow in Combustion Chamber, g/s | $\alpha = \frac{O/F}{(O/F)_{st}}$ | Combustion Chamber Pressure $P_{ch}$, bar | Thrust, N | $\beta_{exp}$, m/s | $\beta_{theor}$, m/s | $\varphi_\beta$ |
|---|---|---|---|---|---|---|---|---|---|
| Pure kerosene experimental results | | | | | | | | | |
| 27.71 | 15.89 | 29.80 | 73.42 | 0.17 | 6.12 | 76 | 944 | 1211 | 0.775 |
| 27.73 | 16.95 | 22.84 | 67.53 | 0.20 | 6.20 | 73 | 1039 | 1251 | 0.830 |
| 29.80 | 13.55 | 24.37 | 67.73 | 0.23 | 6.53 | 79 | 1091 | 1286 | 0.848 |
| 26.52 | 15.34 | 29.75 | 71.62 | 0.17 | 5.93 | 70 | 937 | 1210 | 0.774 |
| 26.69 | 15.37 | 28.74 | 70.80 | 0.17 | 5.93 | 70 | 948 | 1210 | 0.783 |
| 26.62 | 15.72 | 26.36 | 68.71 | 0.18 | 5.87 | 70 | 967 | 1232 | 0.785 |
| 26.74 | 16.10 | 23.99 | 66.84 | 0.195 | 5.86 | 71 | 992 | 1245 | 0.796 |
| 26.84 | 16.54 | 23.97 | 67.36 | 0.194 | 5.93 | 70 | 997 | 1242 | 0.802 |
| 26.86 | 17.27 | 24.00 | 68.14 | 0.191 | 5.85 | 70 | 971 | 1237 | 0.785 |
| 26.80 | 16.34 | 23.77 | 66.92 | 0.196 | 5.87 | 70 | 994 | 1245 | 0.798 |
| 26.77 | 16.46 | 23.24 | 66.48 | 0.197 | 5.79 | 70 | 986 | 1247 | 0.790 |
| 26.86 | 16.54 | 23.97 | 67.38 | 0.194 | 5.79 | 70 | 973 | 1242 | 0.783 |
| PIB kerosene (0.05%) experimental results | | | | | | | | | |
| 22.62 | 17.50 | 26.95 | 67.07 | 0.150 | 5.21 | 62 | 878 | 1177 | 0.746 |
| 26.23 | 16.58 | 23.36 | 66.17 | 0.193 | 5.77 | 68 | 985 | 1230 | 0.801 |
| 26.37 | 17.05 | 23.48 | 66.90 | 0.191 | 5.75 | 69 | 972 | 1229 | 0.792 |
| 26.74 | 16.96 | 23.20 | 66.90 | 0.196 | 5.81 | 70 | 982 | 1234 | 0.796 |
| 27.10 | 17.26 | 20.80 | 65.16 | 0.209 | 6.18 | 74 | 1073 | 1251 | 0.858 |
| 25.06 | 16.60 | 24.31 | 65.97 | 0.180 | 5.66 | 68 | 971 | 1223 | 0.794 |
| 25.04 | 16.99 | 24.05 | 66.08 | 0.179 | 5.66 | 68 | 969 | 1222 | 0.793 |
| 24.91 | 17.28 | 23.88 | 66.07 | 0.178 | 5.65 | 67 | 968 | 1221 | 0.792 |
| 24.79 | 16.60 | 23.78 | 65.17 | 0.181 | 5.54 | 66 | 961 | 1223 | 0.786 |
| 24.83 | 16.96 | 21.77 | 63.56 | 0.189 | 5.62 | 66 | 1000 | 1234 | 0.810 |
| 24.85 | 17.19 | 21.99 | 64.03 | 0.187 | 5.52 | 66 | 976 | 1231 | 0.793 |
| 24.90 | 16.84 | 24.19 | 65.93 | 0.178 | 5.57 | 67 | 955 | 1220 | 0.783 |
| 24.81 | 17.04 | 23.93 | 65.77 | 0.178 | 5.54 | 66 | 953 | 1220 | 0.781 |
| 24.99 | 17.35 | 24.37 | 66.71 | 0.176 | 5.64 | 67 | 956 | 1218 | 0.785 |
| 25.01 | 17.78 | 24.28 | 67.07 | 0.175 | 5.63 | 67 | 950 | 1215 | 0.782 |

The problem of the film cooling massflow effect on the wall temperature was studied in detail in the works of the author and his colleagues earlier. Based on these studies and the experience of experimental work, the massflow for film cooling was selected. Details of these studies are described in [12,13].

The kerosene combustion efficiency was determined from the value of the relation between the experimental and theoretical characteristic velocities, as calculated from equilibrium thermodynamical composition of the combustion products [14]:

$$\varphi_\beta = \frac{\beta_{exp}}{\beta_{theor}} = \frac{\frac{p_{chexp}F_{thexp}}{m_{chexp}}}{\frac{\sqrt{R_{mixTD}T_{mixTD}}}{\sqrt{n}\left(\frac{2}{n+1}\right)^{\frac{n+1}{2(n-1)}}}} \tag{1}$$

where

$\beta_{exp}$—experimental characteristic velocity;

$\beta_{theor}$—theoretical characteristic velocity;

$p_{ch.exp}$—experimental chamber pressure;

$F_{th.exp}$—throat area;

$m_{ch.exp}$—combustion chamber massflow;

$R_{mixTD}$, $T_{mixTD}$, and $n$equilibrium composition gas constant for kerosene and oxygen combustion products, temperature and polytrophic constant computed at experimental combustion chamber pressure, and oxidizer to fuel ratio.

Fire tests were performed under ambient conditions using the Rocket Engines Department test bench at the Moscow Aviation Institute. Pure kerosene and kerosene containing 0.05% PIB by mass were

used during the tests, and the characteristic velocity βexp of the engine was measured. This amount of PIB was selected from the results of the study [9], which showed that the concentration of PIB reduces the hydrodynamic drag in the pipe at the maximum possible value, other parameters being equal. The other measured parameters were fuel massflow, oxidizer massflow, outer wall temperature at 12 points, and combustion chamber pressure.

According to [15], the use of gaseous atomization eliminates the influence of the liquid properties on the linear sizes of the drops, and the droplet diameter is mainly determined by the atomization gas parameters. As the injectors used in the test engine are of the gaseous atomization type (atomized by gaseous oxygen similar to "air-blast atomizers"), it could be stated that the Sauter mean diameter (SMD) under these conditions and using the current design is mainly influenced by the oxidizer massflow and is not dependent on fuel properties such as kinematic viscosity.

The influence of PIB additives on heat exchange inside the combustion chamber and wall film cooling efficiency was estimated with data derived from 12 Type-K thermocouples, that were placed on the outer wall of the combustion chamber in the configuration shown in Figure 1: four thermocouples each in the three combustion camera sections.

## 4. Results

Before conducting the fire tests, several hydraulic tests of the fuel feed system were performed, and the profile of massflow dependence on the pressure drop was established (Figure 2). Figure 2a shows that hydraulic drag through the fuel feed system remained constant up to 30 g/s and pressure drop of 35 bar. A moderate change of the PIB kerosene massflow at an equal pressure drop can be explained by the effect of discharge coefficient increase for flows with enhanced viscosities in swirl injectors. This non-intuitive effect is due to the gaseous vortex diameter reduction, which was observed in [16]. There, the gaseous core appeared to be smaller for higher viscosity liquid, and the massflow increased together with the film thickness when the wall-bounding flows were discussed. Viscosity increases friction losses and the tangential velocity of kerosene in the swirl injector decreases. Therefore, for high liquid viscosities, the air core breaks down, increasing the cross-sectional liquid area, which means higher massflow. Therefore, at the same pressure drop, the kerosene massflow with PIB through the swirl injector will be greater than the massflow of pure kerosene despite the large friction losses caused by the higher viscosity.

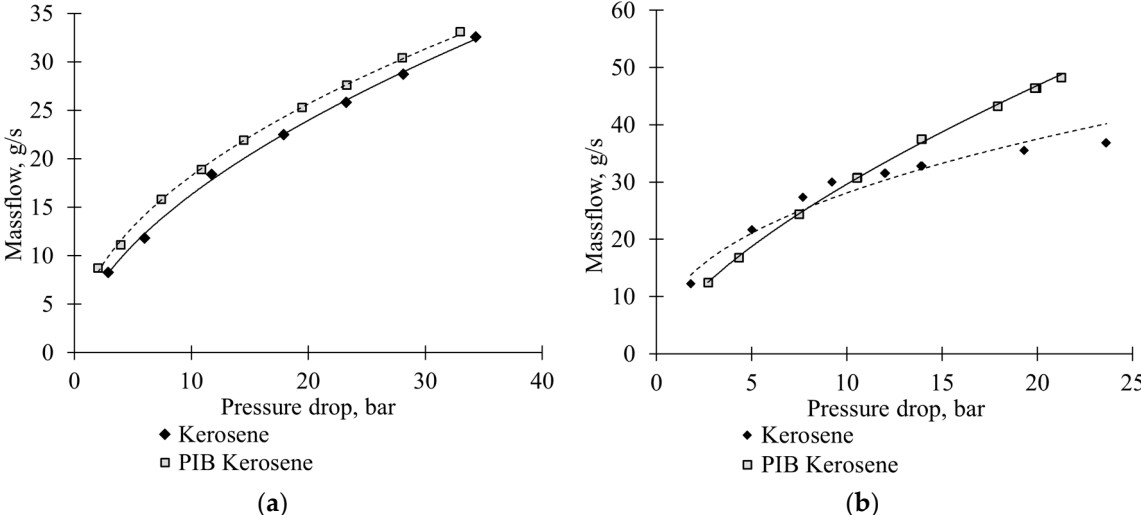

**Figure 2.** Correlation between massflow through the feed system and pressure drop under atmospheric conditions: (**a**) fuel injector massflow; (**b**) fuel film cooling massflow.

The results of the film feed system hydraulic tests presented in Figure 2b show that hydraulic drag in the wall-film cooling feed system significantly decreases by virtue of PIB presence in kerosene for massflows over 30 g/s and pressure drop over 10 bar. At massflow less than 30 g/s and pressure drops less than 10 bar, the PIB kerosene values for drag are less than those for pure kerosene owing to the absence of the effect observed in swirl injectors, as mentioned earlier [16]. Unlike swirl injectors, wall film cooling injectors with the same pressure drop have higher massflow of pure kerosene, despite the fact that its kinematic viscosity is lower. This is because the wall-film cooling injectors were designed as primitive jet injectors and the viscosity increase for the PIB kerosene finally increased the friction losses and decreased the massflow.

The presented hydraulic tests show that the fire tests were carried out with the absence of drag reduction caused by Toms effect because the fuel massflows for the injector head and wall film cooling were 17 g/s and 24 g/s respectively. Therefore, hydraulic resistance reduction caused by Toms effect was not observed for PIB kerosene flowing over the lines of the experimental LPRE during the fire tests.

During the studies, the fire test time of each LPRE was 2 s. Figure 3 shows how combustion chamber pressure and massflows changed during the fire test. The fire test results and conditions are detailed in Table 2.

Figure 4 represents the correlation between the fuel combustion performance and oxidizer-to-fuel equivalence ratio. The curve in Figure 3 shows that the combustion performance of PIB and pure kerosene are very similar to that when the experimental LPRE is operating with pure kerosene. The assumption of PIB kerosene combustion efficiency reduction due to mean Sauter diameter growth was therefore not confirmed. One possible reason for this could be the change in droplet diameter distribution to a more uniform type of the distribution with the presence of PIB in fuel because the polymer additives enhance the droplet sizes and thus the mean diameter. The quantity of large droplets in the total volume of droplets increased along with the intensity of mixing, this making the total flow parameter distribution more uniform throughout the combustion area. Large droplets thus obtained acted as mixing elements, generating additional turbulence intensity and leaving a trace of the evaporated kerosene behind themselves. The presence of large droplets increased the combustion area surface and caused a greater amount of the fuel to evaporate [6].

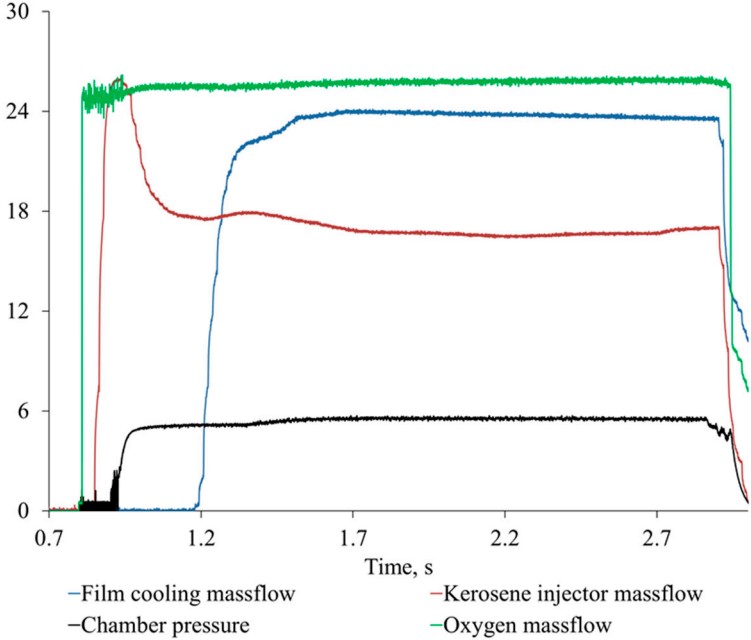

**Figure 3.** Typical for all tests data: combustion chamber, film cooling massflow, kerosene injector massflow, and oxygen massflow.

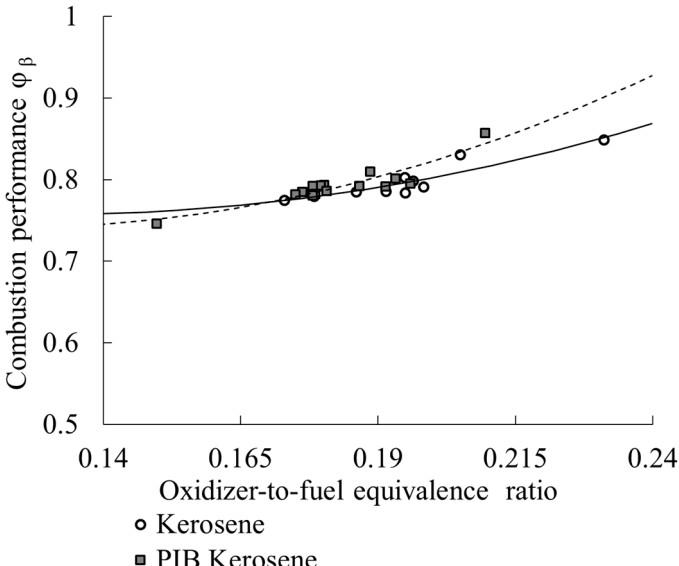

**Figure 4.** Correlation between fuel combustion performance and oxidizer-to-fuel equivalence ratio.

Wall temperature measurements showed lower values for PIB kerosene usage than for pure kerosene. Variation of the wall temperature with time during tests with approximately identical conditions (Experiment No. 10 for pure kerosene and Experiment No. 4 for PIB kerosene) are shown in Figure 5. The experimental LPRE started 1 s after the sensor reading recording commenced, and each experiment lasted 2 s. Massflow values through the injectors and the wall film cooling differed by 1 g/s. The chamber pressure, thrust, total massflow and combustion performance $\varphi_\beta$ were equal for both situations. The maximum wall temperature noted for pure kerosene was 380 °C, whereas for PIB kerosene, it was 250 °C. During the PIB kerosene fire tests the temperature stopped increasing, which indicates a stationary thermal state of the wall. As is commonly known from basic principles, the boiling temperature increases with the increase of ambient pressure. The wall temperature was thus lower than the kerosene boiling temperature at the level of pressure inside the combustion chamber. The phase diagram for kerosene can be found in [17], while a review of the flow features is available in [18]. During the pure kerosene tests, the wall temperature continued to increase gradually in all thermocouples after 2 s. This shows that an evaporated fuel layer near the wall was not yet formed, and the combustion products continued heating the wall surface.

Temperature measurements show that the PIB additive significantly reduces wall temperature, and film cooling is more effective when operating using PIB kerosene. Following the results of the hydraulic tests, it was known that hydraulic drag inside the channels of the injector head increased, which was due to the high viscosity of kerosene with PIB and the influence of the non-Newtonian properties of the liquid. Moreover, a significant drop formation mechanism was present: the co-flowing gaseous flow induced formation of droplets by disintegrating wave crests of the liquid sheet. This follows a common rule: the more viscous the liquid is, the larger are the droplets formed. Beyond this, the droplet size is dependent on the wavelength of the oscillating liquid sheet: smaller drops are formed from shorter waves. Owing to the viscoelastic properties of the non-Newtonian PIB kerosene, the cooling film disintegration slowed down and the wavelength of the sheet increased. This resulted in higher travelling distance of the undistorted liquid sheet [3,19].

Taking into consideration the results of studies [15] that declare no correlation between spraying quality and liquid viscosity in the presence of pneumatic atomization, it can be said that a small enhancement of combustion efficiency with the use of PIB kerosene may be caused by higher efficiency of the wall-film cooling operation. This is due to the wall heat flux reduction degression and combustion chamber heat loss reduction attributed to higher efficiency of wall-film cooling.

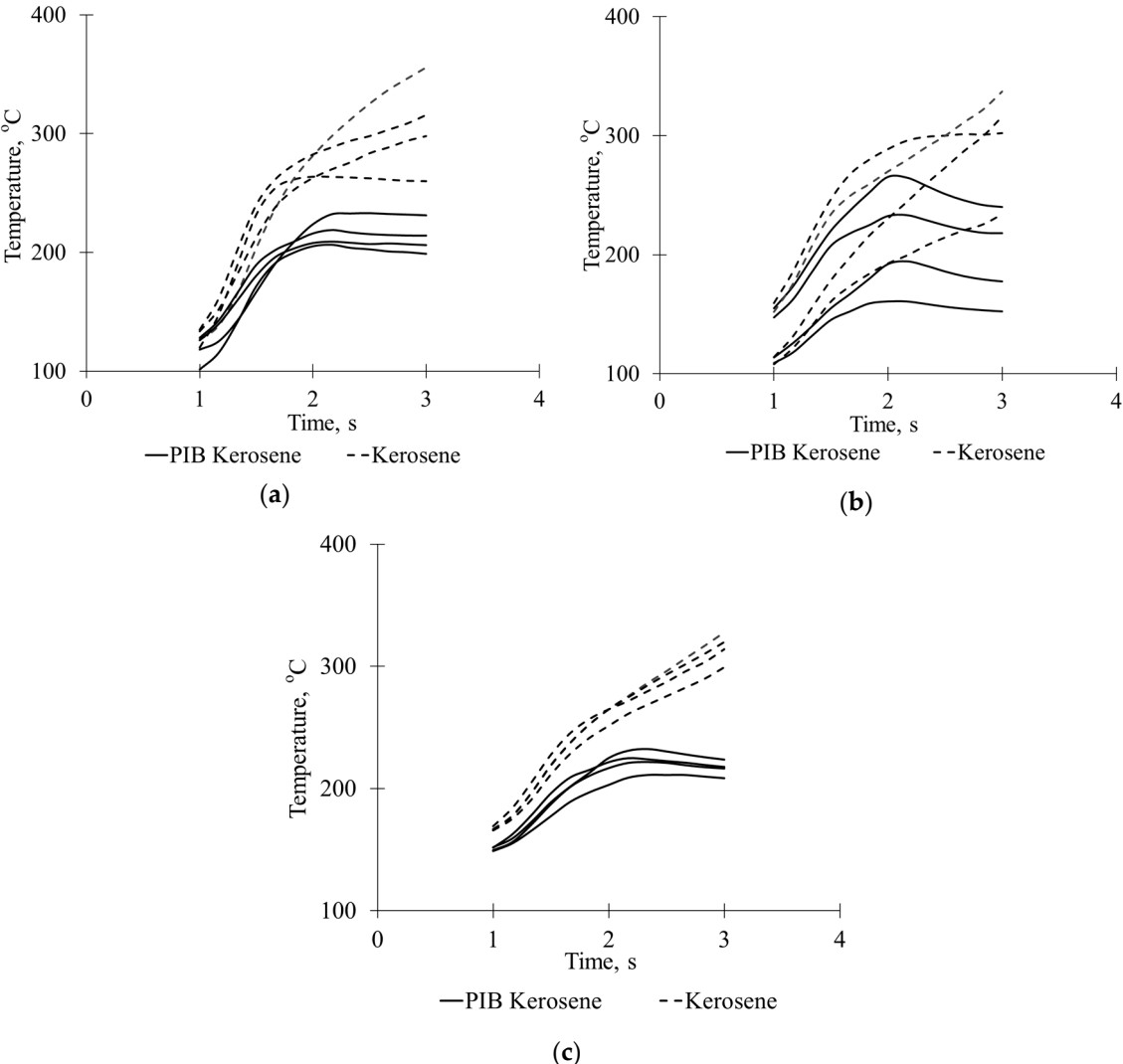

**Figure 5.** Thermocouple data. Thermocouples are set outside the wall for pure kerosene and PIB kerosene tests (see Figure 1): (**a**) Thermocouples 1–4; (**b**) Thermocouples 5–8; (**c**) Thermocouples 9–12.

## 5. Discussion

The study shows that the PIB additive to kerosene increases the LPRE wall-film cooling efficiency. As noted in other similar studies, this research has also not shown any correlation between kerosene PIB additive and combustion efficiency. A small efficiency enhancement with the use of PIB kerosene is obvious due to the higher evaporation rate of the film cooling liquid droplets, thus resulting in higher mixing rates and combustion efficiency.

The results outlined in this paper can help determine the parameters of LPRE combustion chamber cooling systems with the use of PIB additive in kerosene. The combustion chamber cooling system developed with a provision for these effects would allow for specific impulse growth due to decreasing kerosene massflow wall-film cooling. In order to say how much the specific impulse of the combustion chamber will increase, it is necessary to make additional experiments with measuring the temperature of the wall under the conditions typical for combustion chambers of rocket engines with large thrusts. The conditions in the combustion chambers depend on many parameters such as total massflow, oxygen to fuel ratio, design of the injector, design of film cooling nozzles, etc. All these factors will affect the stability of the flow of kerosene film, and therefore the magnitude of the effect of reducing heat flux in wall of the combustion chamber.

Another positive effect of applying PIB additives could be the reduction of the required pump capacity, leading to the decrease in turbine power. As already noted, this is caused by the effect of hydraulic drag decrease in the engine feed line. This may lead to less turbine and gas generator massflow requirements. These changes can bring an augmentation to engine life and reliability, while also enhancing thrust for the same level of fuel reserve in the tanks.

Application of PIB kerosene in LPRE of staged combustion cycle can only insignificantly influence the engine thrust but can lead to reduction of pressure in the gas generator and main combustion chamber. This may enhance the engine reliability and life.

The main effect is estimated to be seen when using the PIB additive in LPRE with gas-generator cycle and especially with fuel-rich gas generator. It is known that a significant proportion of fuel reacts inside the gas generator in this type of LPRE. Combustion products then enter the turbine and outflow to atmosphere without thrust production. In the case of PIB kerosene use, the fuel massflow demand decreases, which means that more fuel can react in the main combustion chamber, leading to the growth in thrust.

Moreover, PIB or other polymer additive application to kerosene or other fuel component can allow for pressure decrease in the fuel supply system of developed and existing LPRE, which can either enhance the engine thrust, or improve its reliability and life, which is of prime importance for recoverable engines and return first stages of launching vehicles.

## 6. Conclusions

In the present work, the combustion efficiency of kerosene with drag reducing polymer (polyisobutylene) has been experimentally studied. Experiments were carried out on a GOX/kerosene rocket engine on a Moscow Aviation Institute test bench. The experiments showed no effect of polyisobutylene on the combustion chamber efficiency. However, the effect of the polyisobutylene addition to kerosene on the combustion chamber wall film cooling efficiency was demonstrated. The wall film cooling efficiency of the PIB kerosene significantly increased in comparison with the pure kerosene. Analysis of the results showed that this effect is apparently associated with the appearance of non-Newtonian viscoelastic properties in kerosene with PIB. The turbulent flow of gases in the combustion chamber does not dilute the PIB kerosene cooling film, which makes it retain its efficiency over a longer wall length. For this reason, the use of PIB kerosene as a fuel in liquid propellant rocket engines can increase the efficiency of combustion chamber cooling, which increases its reliability and reusability.

**Author Contributions:** I.B. and E.S. have designed and investigated the fire tests and are the main authors of the paper. V.G., V.C. and P.L. are the specialists in PIB application in kerosene rocket engine who revised paper critically for important intellectual content. A.K. and Y.Y. participated in the analysis of the data and improvement of the paper.

**Funding:** This research was funded by the Russian Ministry of Education and Science (Project number 13.7418.2017/8.9).

**Conflicts of Interest:** The authors declare no conflict of interest.

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
