# Peer review of "Influence of Polyisobutylene Kerosene Additive on Combustion Efficiency in a Liquid Propellant Rocket Engine"

_aerospace, doi:10.3390/aerospace6120129_

Round 1
Reviewer 1 Report
The paper presents an experimental investigation into the effect of polymer addition in kerosene fuel propellant in rocket engines. The study is using an instrumented test engine where liquid fuel (neat or mixed with additive) can be sprayed into the thrust chamber, as well as fed in the periphery of the chamber for wall/film cooling. The present experiments continue past research by the authors and their research group in the same test engine: here the authors explore the effects of the polymer additive (polyisobutylene) on the overall combustion efficiency, where they conclude that there is little discernible effect, and on the film cooling, where they show a substantial increase in cooling capacity with polymer addition. The experiment is well designed, and the results - both null (combustion efficiency) and positive (film cooling) - are important and worth publication. This is an interesting paper that should be published after minor corrections, mostly explanatory, that are detailed below. The authors should be commented on the 'heroic' nature of their experiments; few people approach this topic with a clear idea towards simplifying the challenging system and understanding the underlying physics.
Line 42: why is the droplet size distribution an important parameter here? You have added references to the effect, but a small explanation of your comment will make the present paper more 'self-sufficient';
Line 50, and many places thereafter: "Toms effect" needs appropriate reference, and a small explanation of what it is. This would enhance your presentation and increase your chances of being cited;
Line 57, and many places thereafter: liquid "disintegration" is what researchers in other fields call "atomization". Many of the arguments that you form can be connected to the literature of, say, diesel sprays, or even jet engines. Again, more visibility and impact would ensue if you tie your results to those
Table 1: "Chilling" temperature?
Table 2: We lack clear definitions here. A reader can guess the meaning but the nomenclature is different in the the "western" vs the "eastern" literature (c*_{th}, c*_{exp}, and eta* are the ones this reviewer is more familiar with). Move the equations from below to the first occurrence of these terms after the table
Table 2: please provide an explanation of the fuel rich condition. Is it mimicking parameters of a large engine, or was there some other reason associated with film cooling, or yet with the chamber temperature?
Line 87: please provide a reference to explain your comment about "largest hydrodynamic drag decrease"
Line 94: more explanation is needed here; are you discussing the oxidizer injectors, and since you are using GOx instead of LOx you do not need to worry about oxidizer atomization, or are you referring to the fuel injectors? Are those using oxidizer gas for liquid fuel atomization ("blast atomizers")? Be more specific, and if need be draw a 2D figure that explains the injector geometry
Line 106: Explain what the gaseous vortex diameter reduction is. Perhaps with a figure/sketch of the process as you envision it. The reader should not have to guess the meaning here (is it the backward-facing step formed by the oxidizer gas injected at the periphery, and the liquid in the center?)
Figure 2: Are the lines fit to the data (power laws, square root per theory, etc.?) Would be best if you provide the mathematical form of the fits, and calculate an equivalent discharge coefficient.
Line 116:Again, explain the comment in more detail. Your results should be independent from previous work; Currently your discussion depends heavily on a close reading of references 12 and 13. Here you could compare your discharge coefficients to those expected in swirl injectors/atomizers (e.g. Sutton)
Line 131/132: F_{th,exp} is better than the F_{th}_{exp} that you use, which can get a little confusing
Line 133: " и " что?
line 170/171: this is a good result, solidly proven. Perhaps you should emphasize that more in your abstract, possibly putting it first, before discussing the "null" result on combustion efficiency
Line 175/176: True, but you can explore this a bit more; give a reference for this "common rule". Kelvin-Helmholtz mechanism, Rayleigh-Taylor, etc., there is plenty of literature in spray formation/atomization that can be used here.
Line 182: Does this statement not contradict the previous statement about "the more viscous the liquid," etc.?
Author Response
The paper presents an experimental investigation into the effect of polymer addition in kerosene fuel propellant in rocket engines. The study is using an instrumented test engine where liquid fuel (neat or mixed with additive) can be sprayed into the thrust chamber, as well as fed in the periphery of the chamber for wall/film cooling. The present experiments continue past research by the authors and their research group in the same test engine: here the authors explore the effects of the polymer additive (polyisobutylene) on the overall combustion efficiency, where they conclude that there is little discernible effect, and on the film cooling, where they show a substantial increase in cooling capacity with polymer addition. The experiment is well designed, and the results - both null (combustion efficiency) and positive (film cooling) - are important and worth publication. This is an interesting paper that should be published after minor corrections, mostly explanatory, that are detailed below. The authors should be commented on the 'heroic' nature of their experiments; few people approach this topic with a clear idea towards simplifying the challenging system and understanding the underlying physics.
Line 42: why is the droplet size distribution an important parameter here? You have added references to the effect, but a small explanation of your comment will make the present paper more 'self-sufficient';
Explanation added in Line 47-52:
This is due to the fact that completely different shapes of the droplet diameter distributions can correspond to the same DMD. Also, two injectors spraying a liquid with the same average diameter can provide different combustion efficiency due to the fact that their design will give different droplet size distributions. This is because a different ratio of large and small droplets injected will give different combustion efficiency [5, 6].
Line 50, and many places thereafter: "Toms effect" needs appropriate reference, and a small explanation of what it is. This would enhance your presentation and increase your chances of being cited;
Explanation added in Line 30-33:
The decrease in hydrodynamic drag is caused by the Toms effect [1]. In 1948, B. Toms recognized the reduction in wall shear stress caused by the addition of small amount of polymer to a turbulent flowing fluid. This effect is widely used in pipeline transportation of oil and shipbuilding to reduce friction.
Line 57, and many places thereafter: liquid "disintegration" is what researchers in other fields call "atomization". Many of the arguments that you form can be connected to the literature of, say, diesel sprays, or even jet engines. Again, more visibility and impact would ensue if you tie your results to those
Fixed in Line 67
Table 1: "Chilling" temperature?
Fixed in Table 1
Table 2: We lack clear definitions here. A reader can guess the meaning but the nomenclature is different in the the "western" vs the "eastern" literature (c*_{th}, c*_{exp}, and eta* are the ones this reviewer is more familiar with). Move the equations from below to the first occurrence of these terms after the table
Fixed in Line 106-117
Table 2: please provide an explanation of the fuel rich condition. Is it mimicking parameters of a large engine, or was there some other reason associated with film cooling, or yet with the chamber temperature?
Explanation added in Line 96-104:
Such test conditions were used to reduce the temperature of the hot gas near the wall and to prevent overheating of the combustion chamber wall. Due to the fact that the liquid film under the combustion chamber conditions is atomized by the hot gas flow, therefore, the massflow of kerosene for the wall film cooling was chosen so that it was guaranteed to prevent this. The problem of the film cooling massflow effect on the wall temperature was studied in detail in the works of the author and his colleagues earlier. Based on these studies and the experience of experimental work, the massflow for film cooling was selected. Details of these studies are described in [12, 13]
Line 87: please provide a reference to explain your comment about "largest hydrodynamic drag decrease"
Explanation added in Line 120-123
This amount of PIB was selected from study [9] showed that this concentration of PIB reduces the hydrodynamic drag in the pipe at the maximum possible value of other parameters being equal.
Line 94: more explanation is needed here; are you discussing the oxidizer injectors, and since you are using GOx instead of LOx you do not need to worry about oxidizer atomization, or are you referring to the fuel injectors? Are those using oxidizer gas for liquid fuel atomization ("blast atomizers")? Be more specific, and if need be draw a 2D figure that explains the injector geometry добавлено уточнение
Explanation added in Line 128
As the injectors used in the test engine are of the gaseous atomization type (atomized by gaseous oxygen similar to “air-blast atomizers”),...
Line 106: Explain what the gaseous vortex diameter reduction is. Perhaps with a figure/sketch of the process as you envision it. The reader should not have to guess the meaning here (is it the backward-facing step formed by the oxidizer gas injected at the periphery, and the liquid in the center?)
Explanation added in Line 140-149
A moderate change of the PIB kerosene massflow at an equal pressure drop can be explained by the effect of discharge coefficient increase for flows with enhanced viscosities in swirl injectors. This non-intuitive effect is due to the gaseous vortex diameter reduction, which was observed in [16]. There, the gaseous core appeared to be smaller for higher viscosity liquid, and the mass flow increased together with the film thickness when the wall-bounding flows were discussed. Viscosity increases friction losses and the tangential velocity of kerosene in the swirl injector decreases. Therefore, for high liquid viscosities, the air core breaks down, increasing the cross-sectional liquid area, which means higher mass flow. Therefore, at the same pressure drop, the kerosene mass flow with PIB through the swirl injector will be greater than the mass flow of pure kerosene despite the large friction losses caused by the higher viscosity.
Figure 2: Are the lines fit to the data (power laws, square root per theory, etc.?) Would be best if you provide the mathematical form of the fits, and calculate an equivalent discharge coefficient.
Trend lines are shown in order to show that in one case, the points are located at almost equal distances, and in the other case they intersect each other. This clearly shows the absence of the viscosity effect for swirl injectors in the second case and its presence in the first case. The mathematical formulation this curve in this case does not carry any value for the reader and is not used or discussed hereinafter. In these figures, it was important to draw the reader's attention to how the experimental points are located relative to each other for the two types of injectors.
Line 116:Again, explain the comment in more detail. Your results should be independent from previous work; Currently your discussion depends heavily on a close reading of references 12 and 13. Here you could compare your discharge coefficients to those expected in swirl injectors/atomizers (e.g. Sutton)
Explanation added in Line 158-162
Unlike swirl injectors, wall film cooling injectors with the same pressure drop have higher massflow of pure kerosene, despite the fact that its kinematic viscosity is lower. This is because the wall-film cooling injectors were designed as primitive jet injectors and the viscosity increase for the PIB kerosene finally increased the friction losses and decreased the mass flow..
Line 131/132: F_{th,exp} is better than the F_{th}_{exp} that you use, which can get a little confusing
Fixed in Line 112-114
Line 133: " и " что?
Fixed in Line 115
line 170/171: this is a good result, solidly proven. Perhaps you should emphasize that more in your abstract, possibly putting it first, before discussing the "null" result on combustion efficiency
Fixed in Line 16
Line 175/176: True, but you can explore this a bit more; give a reference for this "common rule". Kelvin-Helmholtz mechanism, Rayleigh-Taylor, etc., there is plenty of literature in spray formation/atomization that can be used here.
The atomization mechanism of the jet and film is different for each type of non-Newtonian fluids. We do not yet know what type of non-Newtonian fluids kerosene with PIB belongs to. But most likely it refers to viscoelastic fluids like all polymer solutions. Atomization of liquids is studied in this article:
Numerical Models for Viscoelastic Liquid Atomization Spray
Lijuan Qian Jianzhong Lin Fubing Bao
Energies 2016, 9(12), 1079; https://doi.org/10.3390/en9121079
Line 182: Does this statement not contradict the previous statement about "the more viscous the liquid," etc.?
Not. This refers to atomization in air-blast atomizer type injectors.

Reviewer 2 Report
This paper studies the effect of polyisobutylene addition to kerosene on the combustion efficiency and film cooling ability experimentally. Results show virtually no adverse impact on combustion efficiency, whereas the film cooling ability was significantly improved. However, two drawbacks in experimental conditions cause the reviewer to hesitate before recommending this paper for publication; high equivalence ratio and short firing duration. The fuel flow rate for the film cooling is 1.5 to 2 times larger than the primary fuel supply rate, causing the unusually high (fuel-rich) equivalence ratio. Because the firing duration is short, the cooling characteristics in the steady-state condition are not evident. This paper does not provide any history during firings, and the reviewer cannot estimate the impact of initial and shutdown transients. It is challenging to expect improvement in specific impulse on the ground of these experimental results obtained under conditions far from usual design points.
Author Response
Dear Reviewer 2,
This paper studies the effect of polyisobutylene addition to kerosene on the combustion efficiency and film cooling ability experimentally. Results show virtually no adverse impact on combustion efficiency, whereas the film cooling ability was significantly improved. However, two drawbacks in experimental conditions cause the reviewer to hesitate before recommending this paper for publication; high equivalence ratio and short firing duration. The fuel flow rate for the film cooling is 1.5 to 2 times larger than the primary fuel supply rate, causing the unusually high (fuel-rich) equivalence ratio. Because the firing duration is short, the cooling characteristics in the steady-state condition are not evident. This paper does not provide any history during firings, and the reviewer cannot estimate the impact of initial and shutdown transients. It is challenging to expect improvement in specific impulse on the ground of these experimental results obtained under conditions far from usual design points.
_________________________________________________________________________________
However, two drawbacks in experimental conditions cause the reviewer to hesitate before recommending this paper for publication; high equivalence ratio and short firing duration.
Added in Line 96-104
Such test conditions were used to reduce the temperature of the hot gas near the wall and to prevent overheating of the combustion chamber wall. Due to the fact that the liquid film under the combustion chamber conditions is atomized by the hot gas flow, therefore, the massflow of kerosene for the wall film cooling was chosen so that it was guaranteed to prevent this.
The problem of the film cooling massflow effect on the wall temperature was studied in detail in the works of the author and his colleagues earlier. Based on these studies and the experience of experimental work, the massflow for film cooling was selected. Details of these studies are described in [12, 13]:
Alexey Gennadievich VOROBYEV, Svatlana Sergeevna VOROBYEVA, Lihui ZHANG, Evgeniy Nikolaevich BELIAEV, Thermal state calculation of chamber in small thrust liquid rocket engine for steady state pulsed mode, Chinese Journal of Aeronautics, Volume 32, Issue 2, 2019, Pages 253-262 https://doi.org/10.1016/j.cja.2018.12.022.
Vorobiev, A. G., Borovik, I. N., & Ha, S.-U. (2014). Analysis of nonstationary thermal state of low-thrust liquid rocket engine with high-concentration hydrogen peroxide and kerosene propellant with film cooling. VESTNIK of the Samara State Aerospace University, (1(43)), 30. doi:10.18287/1998-6629-2014-0-1(43)-30-40
This paper does not provide any history during firings, and the reviewer cannot estimate the impact of initial and shutdown transients. It is challenging to expect improvement in specific impulse on the ground of these experimental results obtained under conditions far from usual design points.
Firing Test duration 2 seconds sufficient to stabilize the combustion process in the combustion chamber. A new figure 3 has been added to the text of the article showing how pressure and massflow change during the fire test.
Description in detail was also added of the sequence of opening of the fuel valve, oxidizer valve and wall film cooling valve during each test. Test duration of 2 seconds is enough to record a significant decrease in the heating rate of the wall of the combustion chamber. This indicates that the heat flux into the wall has significantly decreased and the wall temperature in the steady-state mode will also be significantly lower.
Added in Line 90-94
“All fire experiments lasted 2 s. In each experiment, the gaseous oxygen valve was the first to open, after 50 ms the spark plug was switched on, after 50 ms the valve of kerosene injectors opened, after that in 200 ms the spark plug was switched off and in 100 ms the film cooling valve was opened. After that, in 1.7 seconds, the valve of the kerosene injectors was closed, then after 30 ms the film cooling valve was closed, and the valve of gaseous oxygen was the last to close.”

Round 2
Reviewer 1 Report
The authors took the reviewer's comments into account, and explained some minor points as requested
Author Response
Dear Reviewer 1,
Thank you.

Reviewer 2 Report
The reviewer understands the adequacy of the test duration. About the validity of expecting improvement in specific impulse on the ground of the present experimental results obtained under conditions far from usual design points, there is no answer. Some explanation about this would be necessary.
Author Response
Dear Reviewer 2
We add some explanation on your comments
"The reviewer understands the adequacy of the test duration. About the validity of expecting improvement in specific impulse on the ground of the present experimental results obtained under conditions far from usual design points, there is no answer. Some explanation about this would be necessary."
in Line 230-236:
" In order to say the specific impulse of the combustor (thruster) increases and determine the value of the increase, it is necessary to make additional experiments including measurements of the wall temperature under the conditions typical for rocket engines with large thrusts. The conditions in the combustors depend on many parameters such as total massflow, oxygen to fuel ratio, design of the injector, design of film cooling nozzles, etc. All these factors will affect the stability of the kerosene film, and therefore the magnitude of the effect of heat flux reduction into the combustor wall"
